# Visualizing the ribonucleoprotein content of single bunyavirus virions reveals more efficient genome packaging in the arthropod host

Erick Bermúdez-Méndez [1,2], Eugene A. Katrukha [3], Cindy M. Spruit[1,4], Jeroen Kortekaas [1,2] & Paul J. Wichgers Schreur [1✉]

Bunyaviruses have a genome that is divided over multiple segments. Genome segmentation complicates the generation of progeny virus, since each newly formed virus particle should preferably contain a full set of genome segments in order to disseminate efficiently within and between hosts. Here, we combine immunofluorescence and fluorescence in situ hybridization techniques to simultaneously visualize bunyavirus progeny virions and their genomic content at single-molecule resolution in the context of singly infected cells. Using Rift Valley fever virus and Schmallenberg virus as prototype tri-segmented bunyaviruses, we show that bunyavirus genome packaging is influenced by the intracellular viral genome content of individual cells, which results in greatly variable packaging efficiencies within a cell population. We further show that bunyavirus genome packaging is more efficient in insect cells compared to mammalian cells and provide new insights on the possibility that incomplete particles may contribute to bunyavirus spread as well.

[1] Department of Virology, Wageningen Bioveterinary Research, Lelystad, The Netherlands. [2] Laboratory of Virology, Wageningen University, Wageningen, The Netherlands. [3] Cell Biology, Department of Biology, Faculty of Science, Utrecht University, Utrecht, The Netherlands. [4]Present address: Department of Chemical Biology & Drug Discovery, Utrecht Institute for Pharmaceutical Sciences, Utrecht University, Utrecht, The Netherlands. ✉email: paul.wichgersschreur@wur.nl

Viruses from the genera *Phlebovirus* (family *Phenuiviridae*) and *Orthobunyavirus* (family *Peribunyaviridae*), belonging to the order *Bunyavirales*, are globally distributed and transmitted between vertebrate hosts by arthropods, such as mosquitoes, sandflies, ticks and midges[1–4]. Several members of these genera cause severe disease in livestock and humans, threatening animal and public health and economies[5,6]. Yet, several fundamental aspects of the viral life cycles remain poorly comprehended.

Phleboviruses and orthobunyaviruses have a tri-segmented genome of single-stranded RNA of negative-sense polarity. The small (S), medium (M) and large (L) segments, named according to their size, are encapsidated by multiple nucleocapsid (N) proteins to form viral ribonucleoprotein (vRNP) complexes that associate with the RNA-dependent RNA polymerase (RdRp or L protein). The N protein is encoded by the S segment, which also encodes a non-structural protein in antigenomic-sense orientation in phleboviruses and in genomic-sense orientation in orthobunyaviruses. The RdRp is encoded by the L segment, whereas the M segment encodes a polyprotein precursor that is cleaved into a non-structural protein and two glycoproteins (Gn and Gc) that protrude from the envelope of mature particles and facilitate entry into host cells[6–8]. Virions are enveloped, spherical particles of ~80–120 nm in diameter[9–11].

From a gene expression perspective, genome segmentation could theoretically facilitate control of viral gene transcription and translation without requiring various *cis*-acting elements as viruses with non-segmented genomes require. Moreover, genome segmentation is generally considered as an evolutionary advantage because it allows genetic reassortment events, which can potentially result in increased viral fitness and transmissibility[12]. However, partitioning of the genome complicates the genome packaging process of segmented viruses, since the packaging of at least one copy of each segment into a particle is thought to be essential to generate infectious progeny. Considering this, it could be expected that the packaging of segmented viral genomes is a highly selective process. The existence of a selective packaging mechanism has already been demonstrated for segmented RNA viruses of other families such as influenza virus and rotavirus[13–15]. Reverse genetics and electron microscopy studies on influenza virus showed that the eight genome segments use packaging signals to assemble into a supramolecular complex with a '7 + 1' configuration[16–19]. Fluorescence spectroscopy combined with pulsed interleaved excitation revealed that rotavirus genome segments form protein-mediated sequence-specific interactions[20]. In both cases, RNA–RNA interactions play an important role in the packaging of the complete genome inside newly formed particles.

Early reports based on mini-genome systems showed that the 5′ and 3′ untranslated regions (UTRs) of bunyavirus RNA segments are directly or indirectly involved in the genome packaging process[21]. Certain flexibility in the packaging process was demonstrated by the rescue of a recombinant Bunyamwera virus (BUNV, genus *Orthobunyavirus*) with an L segment open reading frame flanked by M-type UTRs[22]. Additional work with recombinant viruses revealed the flexibility in the packaging of Rift Valley fever virus (RVFV, genus *Phlebovirus*) genome segments, as evidenced by the creation of multiple two-segmented and four-segmented variants[23–25], as well as a variant with reconfigured coding orientation of the S segment[26]. More recently, by using single-molecule fluorescence in situ hybridization (smFISH) we showed that S, M and L vRNPs of RVFV do not co-localize in the cytoplasm during viral replication. Together with a codon shuffled M segment variant that retained similar growth characteristics, no evidence was found for the formation of a supramolecular RVFV vRNP complex, thereby suggesting that the packaging of RVFV genome segments is not a tightly regulated process[27]. Despite that

the scarce evidence available has provided valuable insights into the genome packaging of bunyaviruses, our understanding of this process is still very limited. In particular, packaging of bunyaviruses has only been studied with a few virus species, and few studies have compared genome packaging in different hosts. Potential host differences in specific steps of the replication cycle may have important implications for virus transmission between vertebrates and invertebrates. In addition, the kinetics and efficiency of generating infectious particles have only been examined at a cell population level and the potential biological role of incomplete particles (i.e., particles lacking one or more genome segments) in within- and between-host transmission is currently unknown.

Here, we use RVFV and Schmallenberg virus (SBV, genus *Orthobunyavirus*) as prototypes of different bunyavirus families to study genome packaging in mammalian and insect cells. We describe a 5-channel FISH-immunofluorescence method that allows simultaneous visualization of progeny virions and each viral genome segment at single-molecule resolution, directly showing that only a small fraction of newly formed virus particles contains a full set of genome segments. We further show at a single-cell level that the packaging efficiency is highly heterogeneous within a cell population and provide direct evidence of the occasional incorporation of antigenomic-sense segments into virus particles. Finally, we report major differences between genome packaging efficiencies in mammalian and insect cells. Thus, the results of this study are in line with our previous suggestion that genome packaging of bunyaviruses is driven by a non-selective process and highlight host cell differences in bunyavirus life cycles.

## Results

**Viral RNA:infectivity ratios differ in mammalian and insect hosts.** To study viral replication and the generation of infectious virus progeny in mammalian and insect cells, we infected Vero E6 (monkey), C6/36 (*Aedes albopictus*) and KC (*Culicoides sonorensis*) cells with RVFV or SBV, quantified in time intracellular and extracellular viral genome segments by RT-qPCR and determined the virus titer in the supernatant by endpoint titration (Fig. 1a). For both RVFV and SBV, the absolute genome segment copy numbers of all three segments were higher in lysates and supernatants of mammalian cells (Vero E6) compared to insect cells (C6/36 and KC) in the logarithmic viral growth phase (Fig. 1b–e). Remarkably, the higher genome copies in supernatants of Vero E6 cells did not always correspond proportionally with higher virus titers. For example, RVFV genome copies obtained at 48 h post infection in Vero E6 cells were more than ten times higher than in C6/36 cells, whereas the virus titers in both host cell lines were equal (Fig. 1f). Another dissonance was observed with SBV at 24 h post infection, where similar genome copies in supernatants of Vero E6 and KC cells resulted in a titer of infectious virus more than ten times higher in KC cells (Fig. 1g). After relating viral RNA copy numbers with virus titers of the supernatants in time, here referred to as vRNA:infectivity ratios, it became clear that for the generation of RVFV and SBV infectious units, fewer genome equivalents are needed in insect cells (Fig. 1h, i), suggesting that bunyavirus genome packaging efficiencies differ between hosts.

In addition to the in vitro comparison between hosts, to gain insight into vRNA:infectivity ratios in vivo, we analyzed plasma samples of lambs experimentally infected with RVFV within the scope of another study[28] (Fig. 1j–l). Briefly, lambs were inoculated via intravenous route with RVFV, followed by daily collection of plasma samples. In these plasma samples, vRNA:infectivity ratios increased over time, with the lowest ratio observed at 2 days post

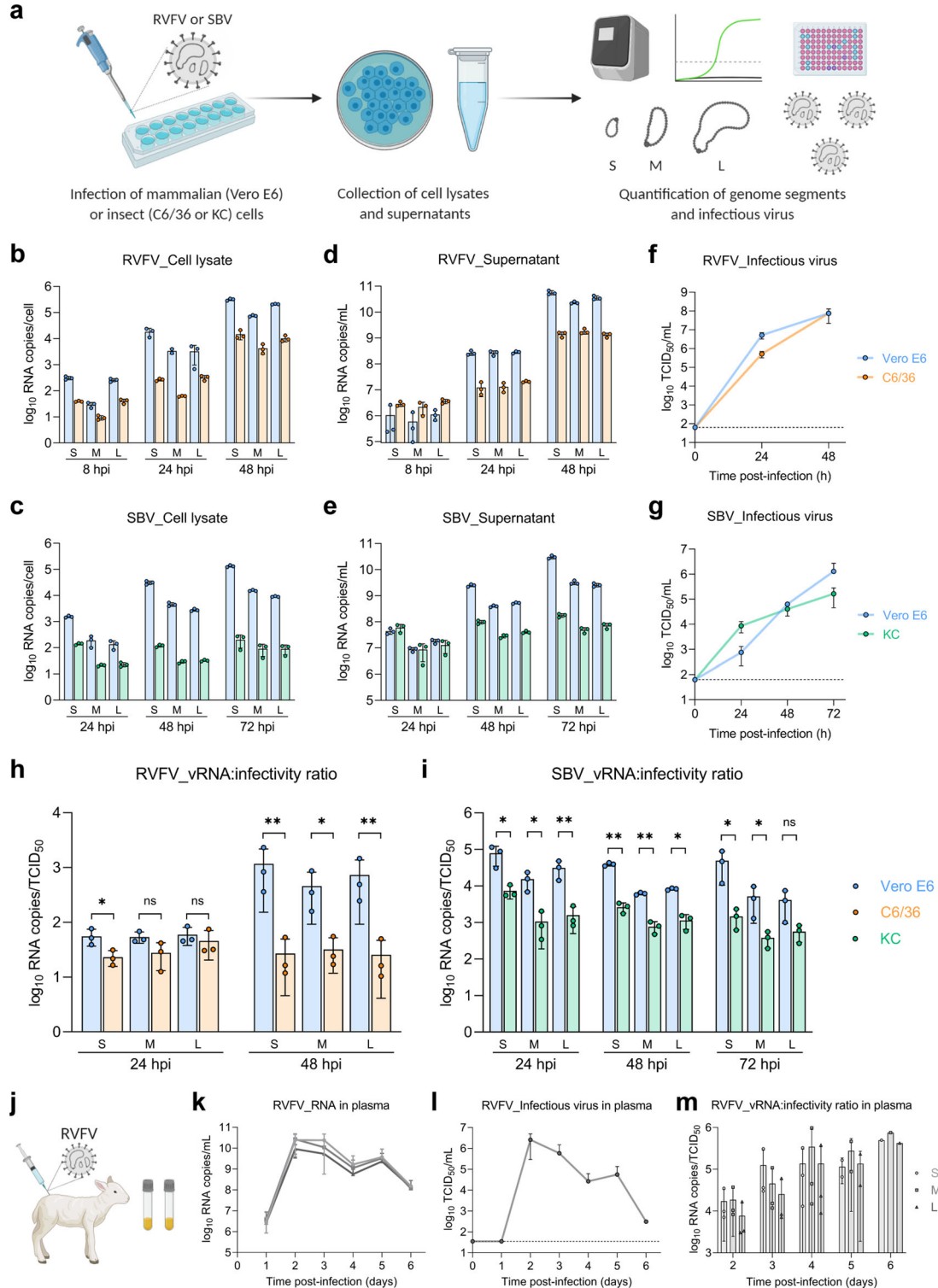

infection (Fig. 1m), coinciding with peak viremia and the onset of symptoms[28], demonstrating that genome packaging efficiencies within a host may differ in time.

**Visualization of newly formed progeny virions at single-particle resolution.** To investigate the release kinetics of progeny virions from infected cells, we developed an immunofluorescence assay using antibodies targeting the surface glycoproteins of RVFV (Gn) and SBV (Gc). We infected Vero E6 cells, fixed the

cells at defined time points and tracked the appearance of virus particles over time (Fig. 2a). For both RVFV and SBV, detection of the glycoproteins became evident around 5 ± 1 h post infection. In the case of RVFV, the Gn glycoprotein signal started to accumulate in a perinuclear region (Fig. 2b, first panel), consistent with the Golgi apparatus being the site of virion assembly[29]. No accumulation of Gc in perinuclear regions was noticed in SBV-infected cells (Fig. 2c). Interestingly, around 7 ± 1 h post infection with RVFV, localized clusters of symmetric spots, most

**Fig. 1 Viral RNA:infectivity ratios in mammalian and insect hosts. a** Schematic representation of the in vitro experimental setup. Mammalian (Vero E6) and insect (C6/36 and KC) cells were infected with RVFV or SBV (MOI 0.01). Cell lysates and supernatants were collected at defined time points. Viral RNA was quantified with genome segment-specific RT-qPCRs and virus titers were determined by endpoint titration. **b**–**e** In vitro replication kinetics of RVFV and SBV. Bars show means with SD. Dots represent biological replicates ($n = 3$ samples). Bar of RVFV cell lysate M segment at 24 h post infection shows mean of two samples. **f**, **g** RVFV and SBV infectious titers in cell culture supernatants. Titers correspond to the same supernatant samples analyzed in **d**, **e**. Graphs show means with SD of $n = 3$ biological replicates. The dashed line indicates the limit of detection ($10^{1.80}$ TCID$_{50}$/mL). **h**, **i** RVFV and SBV vRNA:infectivity ratios calculated as viral genome copies per infectious unit in cell culture supernatants. Bars show means with SD. Dots represent individual ratios ($n = 3$). **j** Schematic representation of the animal samples from another study obtained for analysis. Lambs were experimentally infected via intravenous route with RVFV and plasma samples were collected daily[28]. **k** In vivo replication kinetics of RVFV. Graph shows means with SD of plasma samples ($n = 3$) analyzed by RT-qPCR. **l** RVFV infectious titers in plasma as determined with a virus isolation assay[28]. Graph shows means with SD of plasma samples ($n = 3$). The dashed line indicates the limit of detection ($10^{1.55}$ TCID$_{50}$/mL). **m** RVFV in vivo vRNA:infectivity ratios calculated as viral genome copies per infectious unit in plasma. Bars show means with SD. Dots represent individual ratios ($n = 3$). At early (1 day) and late (5–6 days) times post infection, genome copies and infectious titers of some samples were below the limits of detection. In those cases, the reported values represent the mean of two samples or a single sample. Statistical analysis of vRNA:infectivity ratios was performed using an unpaired two-tailed Student's $t$ test with Welch's correction (not assuming equal variances). *$p < 0.05$; **$p < 0.01$; ns, not significant ($p \geq 0.05$).

likely portraying groups of virus particles trafficking simultaneously from the assembly site to the extracellular space in vesicles, were detected (Fig. 2b, second panel). As the infection progressed, a higher number of virus particles (hundreds to a few thousands) both inside and outside infected cells were detected (Fig. 2b–e and Supplementary Movies 1, 2). Of note, within an infected cell population, several cells showed lower glycoprotein levels and numbers of progeny virions despite being fixed at the same time point, probably representing the intrinsic variability in infection kinetics between cells. A plot of fluorescence intensities of individual spots shows a unimodal intensity distribution characteristic of single particles (Fig. 2f). Likewise, a histogram of the area of the spots also shows a unimodal distribution, denoting reproducible measurements of single spots within and between images (Fig. 2g). Importantly, the single-particle detection of newly formed progeny virions not only allowed us to investigate the kinetics of virion release but also enabled us to determine the genomic composition of individual progeny virions.

**Genome composition of newly formed virus particles**. To investigate the genome content of newly formed RVFV and SBV virions in infected cells, we developed a 5-channel based combined RNA FISH-immunofluorescence method that allows the simultaneous visualization of virus particles and each viral genome segment at single-molecule resolution (Fig. 3a). Virions were detected as described in Fig. 2 and specific FISH probe sets were designed to recognize the S, M and L viral RNAs (Supplementary Figs. 1, 2 and Supplementary Data 1). The method enables the concomitant assessment of viral replication by quantification of the vRNPs in the cytoplasm (Fig. 3d and Supplementary Fig. 3), as well as the determination of the genome content of newly formed virus particles through co-localization analysis between the virions and the vRNPs (Fig. 3b, c, e, f, Supplementary Figs. 3, 4 and Supplementary Movies 3, 4). Importantly, our assay facilitates linking the genomic content of the virions with the cytoplasmic vRNP content of the originative cell.

We used the assay to analyze individual RVFV- and SBV-infected cells fixed at 8 h post infection, a stage in the infection cycle at which release of mature virions is clearly evident (Fig. 3b, c, e, f and Supplementary Figs. 3, 4) and virus genome replication has not proceeded long enough to impede the quantification of vRNPs in the cytoplasm due to an overcrowded signal detection (Fig. 3d and Supplementary Fig. 3). Following analysis, most RVFV and SBV particles were found to be empty, accounting on average for ~55% and 35% of total virions, respectively. In addition, the fraction of particles containing one segment was between ~30–35%, and the fraction containing two segments between about 10 and 20%. The fraction of particles containing a

complete genomic set was below 10% (Fig. 4c). Remarkably, we observed great variability in packaging efficiencies within RVFV- and SBV-infected cell populations. Within both cell populations, a subpopulation of cells showed a striking inefficient packaging process, in some cases seemingly without generating a single infectious particle, whereas other cell subpopulations generated two or more times higher percentages of particles containing a complete genomic set than the average (Fig. 4c). Although the genome packaging process for both viruses is overall inefficient, packaging of SBV genome segments does occur more efficiently than for RVFV.

**Intracellular vRNP content correlates with genome packaging efficiency**. Seeking for an explanation to the high variability in genome packaging efficiency within cell populations, we looked into the vRNP content in the cytoplasm of the individual cells. Quantification of RVFV vRNPs in infected mammalian cells not only exposed a highly heterogenous cell-to-cell composition, but also an overall imbalanced content leaning towards higher abundances of the S (42%) and L (34%) segments compared to the M segment (24%). Quantification of SBV vRNPs in infected mammalian cells demonstrated that the overall vRNP content of the cytoplasm approached a theoretical balance, with abundances near the 33% for all three genome segments. Although the cytoplasm of some SBV-infected cells deviated from the average composition, the cell-to-cell heterogeneity in this population was less pronounced (Fig. 4a, b).

Next, we evaluated whether an imbalanced cytoplasmic content could be associated with a particular genome composition of the virions. The correlation analysis made evident that indeed, if a specific genome segment is more abundant intracellularly, it will be incorporated into a virus particle more often, and vice versa. A strong positive correlation (Pearson's coefficients of at least 0.5660 and $p < 0.01$) was found for all three genome segments of RVFV and SBV (Fig. 4d). The association between the cytoplasmic content and the efficiency of incorporating genome segments into virions was further assessed in a more integrative manner. Based on the frequencies of all three genome segments in the cytoplasm of individual cells and the fractions of empty, incomplete and complete particles, we generated a system to score the balance of the intracellular contents as well as the efficiency of genome packaging, normalizing the scores using the extreme values present in our data set as reference (Fig. 4e). Surprisingly, our analysis revealed that a considerable number of cells with balanced intracellular genome contents exhibited an overall inefficient packaging. This indicates that, although the three different vRNPs will most likely be incorporated into particles in similar numbers if their

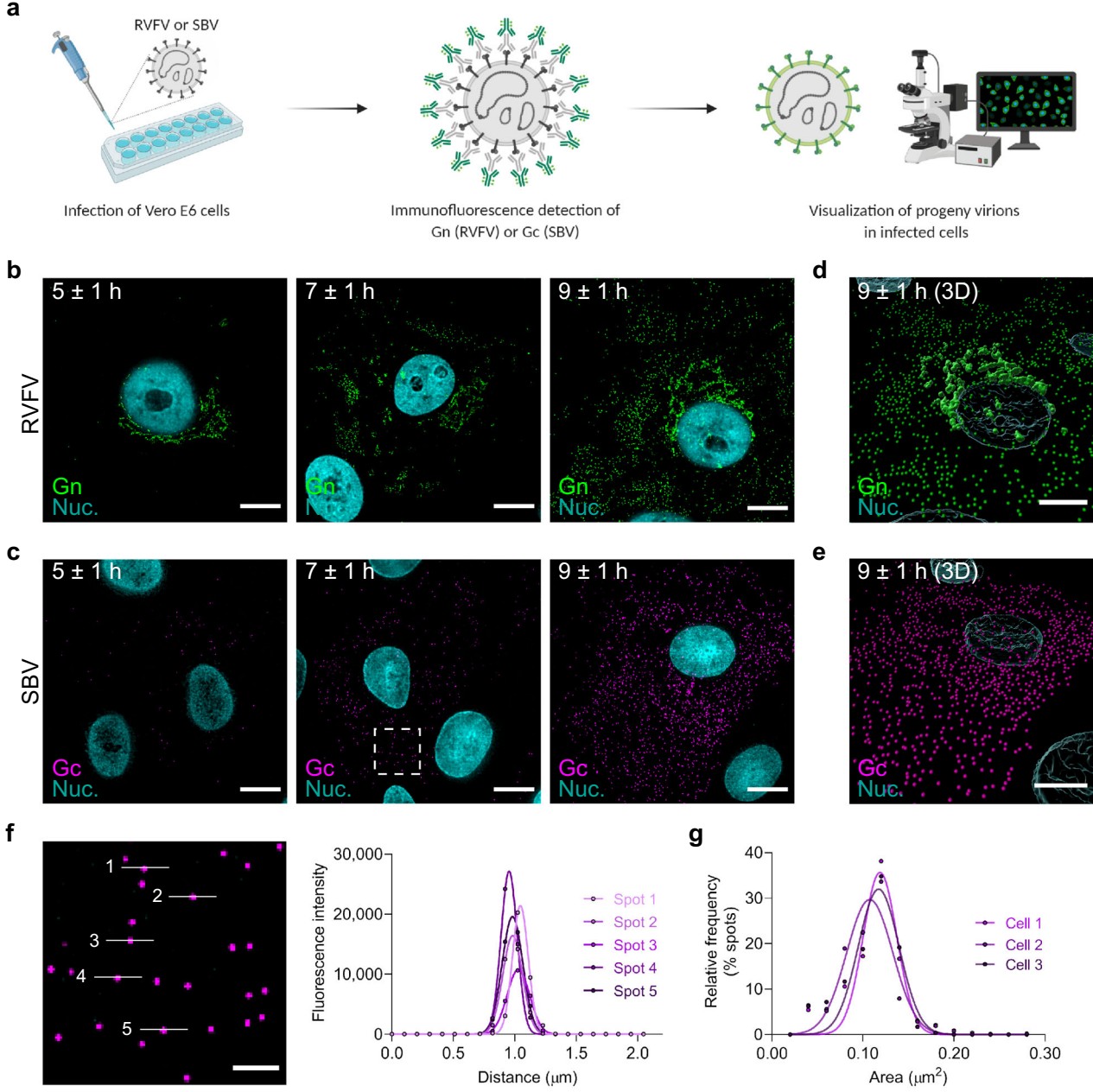

**Fig. 2 Immunofluorescence detection of newly formed bunyavirus progeny virions at single-particle resolution. a** Schematic representation of the experimental setup. Vero E6 cells were infected with RVFV (MOI 1) or SBV (MOI 0.33) and cells were fixed at defined time points. Progeny virions were detected by immunofluorescence. Release kinetics of RVFV particles (green) (**b**) and SBV particles (magenta) (**c**). RVFV virions were detected with antibody 4-D4[48] targeting the Gn glycoprotein in combination with Alexa Fluor 488-conjugated secondary antibodies. SBV virions were detected with serum from an immunized rabbit[52] targeting the Gc glycoprotein in combination with FITC-conjugated secondary antibodies. Cell nuclei (cyan) were visualized with DAPI. RVFV Gn accumulates in a perinuclear region, the site of virion assembly. **d, e** Three-dimensional representations showing the spatial distribution of virions at the 9 ± 1 time point created with Imaris using the Surfaces and Spots modes. **f** Magnification of a region of interest (indicated as a dashed box in the second panel of **c**) and fluorescence intensity plot of the indicated spots. Dots represent data points and lines show Gaussian curves fitting the data. The unimodal distribution of fluorescence intensities along the lines crossing the spots is characteristic of single particles. **g** Histogram of the area of the spots detected in images of SBV-infected cells ($n = 3$ cells; more than 500 spots per image). Dots represent data points and lines show Gaussian curves fitting the data. The unimodal distribution denotes reproducible measurements of single spots within and between images. Images are merged maximum intensity projections of two channels. Scale bars, 10 µm (**b–e**), 2 µm (**f**).

intracellular abundance is similar, the three vRNPs are not necessarily co-packaged into the same particle. However, when we observed relatively efficient packaging, the vRNP content in the cytoplasm was balanced, implying that a balanced intracellular vRNP content is a pre-requisite for relatively efficient genome packaging. Accordingly, it also became clear that an imbalanced cytoplasmic vRNP content generally leads to inefficient genome packaging (Fig. 4e, f).

**Differences in genome packaging efficiencies between mammalian and insect cells.** Based on the different vRNA:infectivity

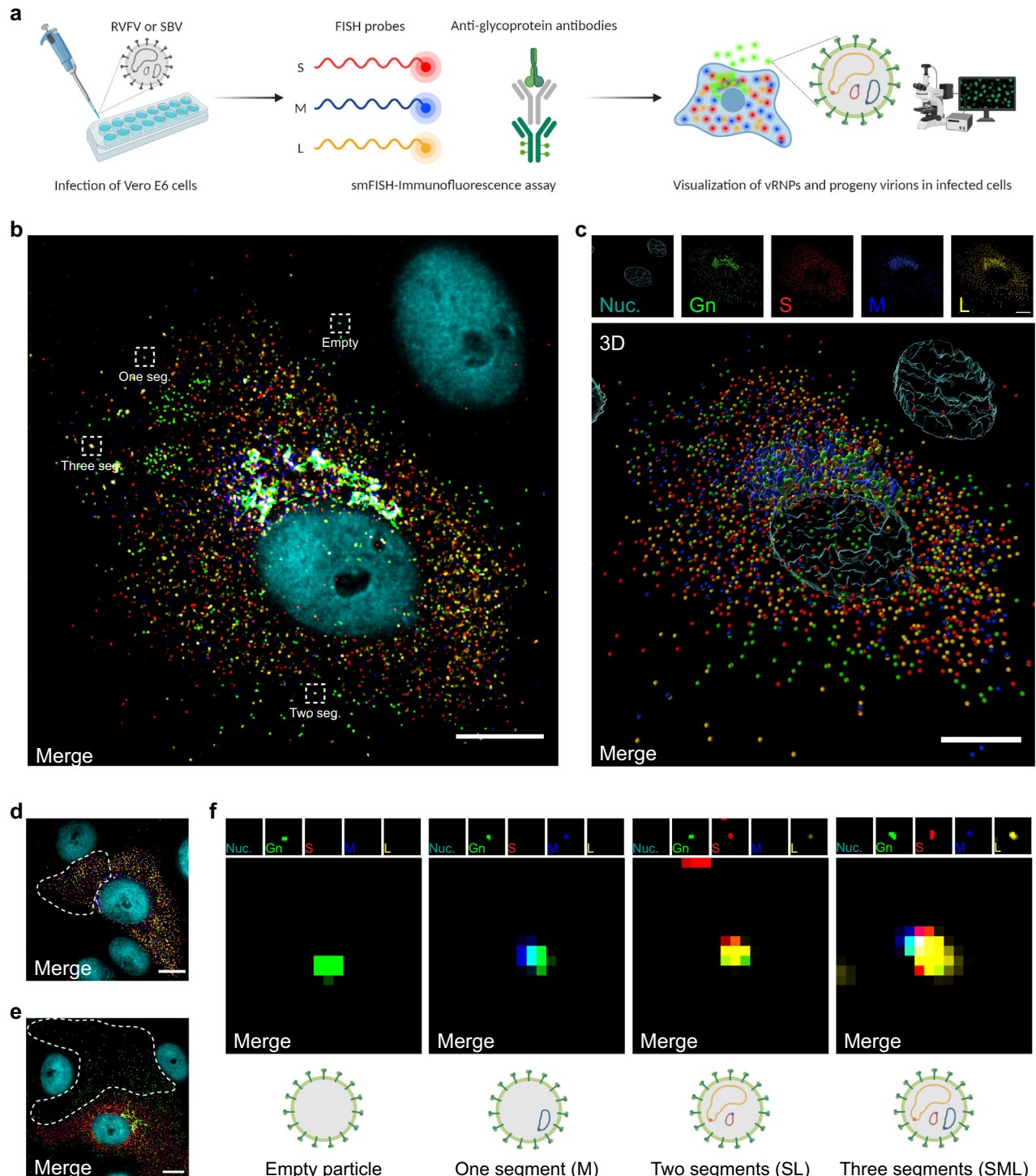

ratios we found between mammalian and insect cells (Fig. 1h, i), we aimed to further evaluate potential host cell differences in genome packaging using our vRNA FISH-immunofluorescence method on RVFV-infected insect cells. Although we managed to visualize RVFV virions and vRNPs in insect cells (Supplementary Fig. 5), the image acquisition and analysis process at single-molecule resolution proved to be very challenging due to the elongated distribution in the z-axis of virion assembly sites. As an alternative, we applied our method to immobilized virions from virus stocks produced in different host cells and compared the genome composition of their virions (Fig. 5a–f). In general, virus stocks consist of a heterogeneous population of empty virions and

virions with one, two or three genome segments (Fig. 5e). Interestingly, in mammalian cells (Vero E6) the S segment was packaged more often than the M and L segments, whereas in insect cells (C6/36) we observed the opposite (Fig. 5f). Consistent with the analysis of newly formed virions (Fig. 4c) and our own previous report[27], about 50% of total RVFV particles produced on Vero E6 cells were empty. On the other hand, empty particles of virus stocks produced on C6/36 cells accounted for a considerably lower fraction (~30% of total virions), indicating that despite bunyavirus genome packaging seems to be a largely stochastic process, the incorporation of genome segments into virions occurs more efficiently in insect cells than in mammalian cells.

**Fig. 3 Single-molecule vRNA FISH-immunofluorescence of bunyavirus infected mammalian cells. a** Schematic representation of the experimental setup. Vero E6 cells were infected with RVFV (MOI 0.50–0.75) or SBV (MOI 0.33) and cells were fixed at 8 h post infection. The S segment (N gene; red), M segment (polyprotein gene; blue) and L segment (RdRp gene; yellow) were hybridized using probe sets labeled with CAL Fluor Red 610, Quasar 670 and Quasar 570, respectively. Progeny RVFV particles (green) were detected with antibody 4-D4[48] targeting the Gn glycoprotein in combination with Alexa Fluor 488-conjugated secondary antibodies. Progeny SBV particles (Supplementary Figs. 3b, 4) were detected with serum from an immunized rabbit[52] targeting the Gc glycoprotein in combination with FITC-conjugated secondary antibodies. Cell nuclei (cyan) were visualized with DAPI. Individual spots, each representing either a single vRNP or a virus particle were detected, counted and assessed for co-localization in ImageJ with the plugin ComDet. **b** Visualization of vRNPs and progeny virions of a RVFV-infected cell. The dashed boxes highlight individual virus particles subjected to co-localization analysis for example purposes. The number of RVFV genome segments in each highlighted particle is indicated. **c** Three-dimensional representation showing the spatial distribution of vRNPs and virions of the image displayed in **b** created with Imaris using the Surfaces and Spots modes. Accumulation of vRNPs and co-localization to the same perinuclear region as Gn show active vRNP recruitment to the site of virion assembly. Co-localization of vRNPs and virions is depicted by merged spheres. **d, e** RVFV-infected cells. The dashed contours represent example regions of interest selected for the quantification of cytoplasmic vRNPs (**d**) and determining the genome composition of extracellular virions through co-localization analysis (**e**) (Supplementary Fig. 3a). Example regions of interest selected for the analysis of SBV-infected cells are shown in Supplementary Fig. 3b. **f** Magnification of regions of interest indicated by dashed boxes in **b**. The genome composition of each virion can be deduced from the spots detected on each individual channel. Images are merged maximum intensity projections of four (**d**) or five (**b, c, e, f**) channels. Due to a higher fluorescence intensity of the green channel compared to the other channels, spots co-localizing with the glycoprotein may sometimes appear masked and not entirely evident in merged images. Scale bars, 10 µm.

In addition, in insect cells the three different genome segments were incorporated into the same virion around three times more often than in mammalian cells (~23% vs. ~7%) (Fig. 5e), generating a higher percentage of complete particles and showing an overall more efficient genome packaging process.

**Visualization of viral complementary RNAs incorporated into newly formed progeny virions**. Previous reports have found viral antigenomes, together with mRNA transcripts here referred to as viral complementary RNAs (cRNAs), in supernatants of bunyavirus infected cells or in purified virions preparations, as evidence for their incorporation into virus particles[26,30–32]. Here, we designed FISH probe sets to specifically recognize the cRNAs of RVFV and directly visualized their packaging using the vRNA FISH-immunofluorescence method (Fig. 6a, Supplementary Fig. 6 and Supplementary Data 1). Due to a maximum capacity to properly filter light wavelengths up to five different channels, we assessed the packaging of one viral segment and the corresponding cRNA in pairs. Indeed, all three RVFV cRNAs were occasionally incorporated into virions (Fig. 6b). Interestingly, we again observed high cell-to-cell variability in packaging efficiency within the cell populations. Furthermore, the ratios between the frequencies of incorporation of the viral genomes and the respective cRNAs differed per segment, resulting in ratios of ~4:1, 9:1 and 14:1 for S/cS, M/cM and L/cL, respectively (Fig. 6c). Although the packaging of cRNAs occurs less frequently than that of viral genome segments, the direct visualization of virions containing cRNAs provides additional evidence of the absence of a selective mechanism that favors exclusively the incorporation of viral genome segments.

## Discussion

The molecular mechanisms involved in the production of infectious bunyavirus progeny are yet to be discovered. Remarkably little is known about the principles that drive the genome packaging process of the multi-segmented bunyavirus genome into virions. Here, we combined smFISH and immunofluorescence assays to determine the genomic composition of RVFV and SBV virions at single-particle resolution by simultaneous detection of individual virus particles and vRNPs (Figs. 2, 3 and Supplementary Figs. 3, 4). Notably, we were able to link the intracellular abundance of specific vRNPs with the composition of progeny virions in individual infected cells and were able to show striking differences between genome packaging efficiencies in mammalian and insect cells.

By analyzing individual infected cells and their progeny virions, we not only observed a high cell-to-cell variability in packaging efficiency, which leads to a highly diverse composition of the progeny virion population, but also learned that the relative intracellular abundance of the vRNPs can influence, at least partially, overall genome packaging efficiencies (Fig. 4b, c). Our observations, obtained from single-cell analysis, are consistent with previous reports on purified virions of RVFV studied at a population level by Northern blotting, which suggested that the relative abundance of each genome segment in virions roughly approximated their relative abundances intracellularly[33,34]. Likewise, we found that a low relative intracellular abundance of a particular genome segment correlates with a low packaging frequency of that segment and vice versa. Accordingly, when the overall intracellular vRNP content was imbalanced (i.e., S:M:L ratio moved away from the theoretical 1:1:1 ratio), virions produced from that cell rarely contained the three genome segments and packaging was most likely very inefficient (Fig. 4d). On the other hand, a balanced intracellular vRNP content appears to serve as an essential precondition for the generation of complete particles, although it does not ensure in all cases an overall efficient genome packaging (Fig. 4e, f). It is worth noting that an imbalanced intracellular vRNP composition can be the consequence of multiple factors, such as differential replication kinetics between the genome segments or an initially imbalanced co-infection of the same cell by a combination of complete and incomplete particles.

Contrary to other segmented RNA viruses like influenza virus and rotavirus, in which specific RNA–RNA interactions facilitate co-packaging of all the different viral genome segments[20,35,36], a growing body of evidence supports the notion that bunyavirus genome packaging is rather flexible and non-selective[21]. Here, we show that less than 10% of RVFV and SBV progeny virions produced in mammalian cells contain the three genome segments, meaning that only a minor fraction of produced virus particles are infectious on their own (Fig. 4c). These results are in line with our previous report[27], which suggested that bunyavirus genome packaging occurs without a specific mechanism that guarantees a consistent incorporation of all three genome segments into the same particle. In addition, we showed that the incorporation of S, M and L cRNAs into virions does occur, but not frequently. Importantly, packaging of cRNAs occurs disregarding whether the corresponding vRNA segment has or not an ambisense coding strategy (Fig. 6b, c). Although we observed similar non-selective features regarding genome packaging of tri-segmented bunyaviruses that belong to two different families, the

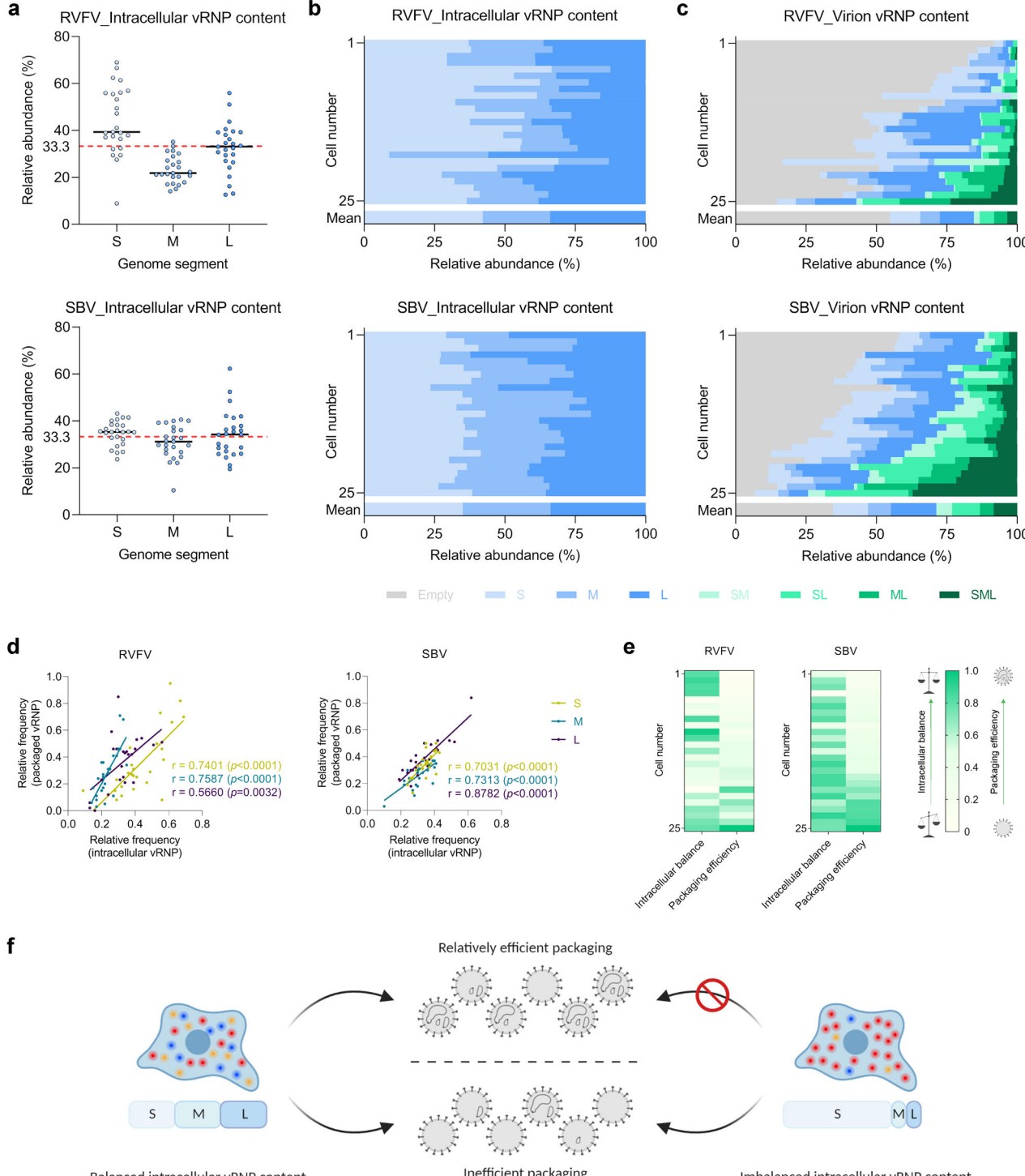

low particle-to-PFU ratios previously reported for BUNV[22] and Crimean-Congo Hemorrhagic fever virus[37] (family *Nairoviridae*, genus *Orthonairovirus*) imply that other bunyavirus species may have evolved towards a more efficient packaging process, but this remains to be studied.

Phleboviruses and orthobunyaviruses sustain a life cycle characterized by alternating productive infections between vertebrates and arthropod vectors[4], underscoring the importance of studying the virus biology in both hosts. In an experimental infection study in goats, the source of the virus was found to cause differences in the course of infection. Insect cell-derived RVFV appeared to be more infectious than mammalian cell-derived RVFV based on

faster peak viremia, infection of peripheral blood mononuclear cells, induction of fever and cytokine levels[38]. From our in vitro virus replication experiments, we noticed that insect cells required fewer genome equivalents per infectious unit compared to mammalian cells (Fig. 1h, i). Furthermore, we found that in RVFV progeny derived from insect cells, the relative amount of particles containing a full set of genome segments were about three times more compared to mature RVFV virions produced in mammalian cells (Fig. 5e). These observations strongly suggest that genome packaging occurs more efficiently in insect cells, which possibly contributes to maintain high viral loads during replication in the arthropod vector to enable efficient transmission to vertebrates.

**Fig. 4 vRNP composition of the cytoplasm of bunyavirus infected mammalian (Vero E6) cells and their progeny virions at a single-cell level.** RVFV- and SBV-infected cells were analyzed with a single-molecule vRNA FISH-immunofluorescence method as described in Fig. 3. **a** Quantification of RVFV and SBV S, M and L vRNPs in the cytoplasm of infected cells. Data are expressed as the relative intracellular abundance of each vRNP. The black lines represent the medians ($n = 25$ cells) and the red dotted line represents a theoretically balanced abundance of 33.33%. **b** Data shown in **a** presented per individual cell. **c** Quantification of RVFV and SBV S, M and L vRNPs in progeny virions. Data are expressed as the relative abundance of each of the eight different potential compositions of virions. Graphs **b** and **c** show the composition results of single cells ($n = 25$ cells; more than 5000 RVFV virions and more than 4500 SBV virions) and means. Cell numbers in **b** and **c** correspond. **d** Correlation analysis between the relative intracellular frequency of a specific genome segment and the relative frequency of that genome segment being packaged. Pearson's correlation coefficients ($r$) and $p$ values are shown for each genome segment. **e** Relationship between the intracellular content of vRNPs and the packaging efficiency of individual cells. A generic system to score the intracellular balance and the packaging efficiency was created. A frequency of 0.33 for each genome segment was considered as theoretically balanced. The balance score was calculated as the summatory of the absolute deviations from the theoretical frequency, normalized from 0 to 1, assigning the least balanced composition of the data set a score of 0. The packaging efficiency score was calculated taking into account the frequency of empty, incomplete and complete virus particles, normalized from 0 to 1, assigning the most efficient packaging value of the data set a score of 1. Scores are color coded from light green (lowest) to dark green (highest). **f** Proposed model on the efficiency of genome packaging based on the intracellular vRNP content. A balanced vRNP content in the cytoplasm seems to be a pre-requisite for relatively efficient genome packaging.

The reasons behind the more efficient genome packaging in insect cells are yet unknown, but could be related to the evolutionary origin of the viruses, which has been suggested to be of arthropod-specific ancestors[39].

In addition to host differences, the fact that the vRNA:infectivity ratio in plasma samples from experimentally infected lambs increased over time indicates that genome packaging efficiency may vary within a single host over the course of infection (Fig. 1m). To better evaluate changes in packaging efficiency over time, analysis of the vRNP content of virions present in plasma would be very informative. However, immobilizing virions from the plasma matrix has proven to be technically very challenging. Alternatively, single-cell analysis of vRNP packaging efficiencies at time points later than those evaluated in the present study could provide relevant information. Unfortunately, at a later stage of infection, such analysis is hampered by the increased intracellular vRNP density, which results in accumulated signal throughout the cytoplasm leading to loss of single-molecule resolution.

Interestingly, the bunyavirus genome packaging process investigated here gives rise to a large fraction of incomplete virus particles lacking one or two genome segments (Figs. 4c and 5e). Recently, a study with an influenza virus strictly dependent on genome complementation by co-infection demonstrated that incomplete influenza virus particles contributed to localized within-host spread[40]. This raises the intriguing question of whether co-infection by complementing incomplete particles may compensate for the inefficiency observed in bunyavirus genome packaging. In this hypothetical scenario, where complete particles are dispensable for a productive infection, bunyaviruses may resemble the life cycle of multipartite viruses, which establish a productive infection by independent transmission of a complementary ensemble of particles each containing a single genome segment[41,42].

Besides the potential role that incomplete particles may play in dissemination of bunyaviruses, additional strategies that would increase the flexibility also seem plausible as ways to overcome the bottleneck of an overall inefficient genome packaging process. Incorporating more than three genome segments per particle increases the probability of packaging at least one copy of S, M and L segments. Cryo-electron microscopy analyses of RVFV particles[43,44] suggest that additional genome segments would fit within the intra-virion space. Another potential strategy involves the transmission of a large number of virions in structures known as collective infectious units, which result in a locally increased multiplicity of infection (MOI)[45]. It should also be noted, that a flexible packaging process may actually be best suited for the changing environments faced by the virus during its life cycle between vertebrates and arthropods. Finally, flexible packaging capabilities in terms of non-selectivity towards specific RNA sequences also facilitate the occurrence of reassortment events with related viruses, which increases genetic diversity and favors virus evolution.

In summary, here we studied genome replication and packaging of prototype bunyaviruses in mammalian and insect cells, both at a single-particle and single-cell level, as well as at a virion population and cell population level. Taken together, the evidence presented in this report further demonstrates that packaging of bunyavirus genome segments is a flexible, non-selective process and that genome packaging is more efficient in insect cells compared to mammalian cells.

## Methods

**Cell lines.** Vero E6 cells (ATCC CRL-1586) were maintained in minimum essential medium (MEM) supplemented with 5% fetal bovine serum (FBS), 1% antibiotic/antimycotic, 1% MEM non-essential amino acids (MEM NEAA) and 2 mM L-glutamine at 37 °C and 5% CO$_2$. C6/36 cells (ATCC CRL-1660) were maintained in L-15 medium (Leibovitz) (Sigma-Aldrich) supplemented with 10% FBS, 1% antibiotic/antimycotic, 1% MEM NEAA and 2% tryptose phosphate broth at 28 °C. KC cells were maintained in Schneider's *Drosophila* medium supplemented with 10% FBS and 1% antibiotic/antimycotic at 28 °C. Cell culture media and supplements were purchased from Gibco, unless specified otherwise.

**Viruses.** Virus stocks of RVFV strain Clone 13[46] were obtained after infection of Vero E6 or C6/36 cells at a MOI of 0.005. Virus stocks of SBV isolate NL-F6[47] were obtained after infection of Vero E6 cells at a MOI of 0.01.

**Genome segment-specific quantitative RT-PCR.** Mammalian cells (Vero E6) or insect cells (C6/36 for RVFV and KC for SBV) were seeded in 6-well cell culture plates at $2 \times 10^5$ cells/well or $6 \times 10^5$ cells/well, respectively, and allowed to attach for 2–4 h. Cells were subsequently infected at a MOI of 0.01 and after incubation for 3.5 h, the inoculum was removed and substituted with fresh medium. At defined time points (varied per experiment), samples from the culture supernatant and cells were collected. In addition to the in vitro experiments, plasma samples were obtained from another study (lambs #158, #160 and #162) in which lambs were experimentally infected via intravenous route with a $10^5$ tissue culture infectious dose (TCID)$_{50}$ dose of RVFV strain 35/74[28].

From 1–2 mL of cell lysate, 200 μL of culture supernatant or 200 μL of plasma, total nucleic acid extractions were performed with the NucliSENS easyMAG system (bioMérieux) according to the manufacturer's instructions. Subsequently, viral cDNA was synthesized with the SuperScript IV First-Strand Synthesis System for RT-PCR (Invitrogen) using a combination of S, M and L segment-specific primers (Supplementary Table 1), according to the manufacturer's instructions. After the reverse transcription reaction, quantitative PCR amplifications were performed with the Power SYBR Green PCR Master Mix using 5 μL of 20- or 200-fold diluted cDNA preparations in a total volume of 25 μL, in combination with a 7500 Fast Real-Time PCR System (Applied Biosystems). Fragments from each segment were amplified using specific primers (Supplementary Table 2) under the following conditions: an initial denaturation step at 95 °C for 10 min; 40 cycles of denaturation at 95 °C for 15 s, annealing at 59 °C for 30 s and extension at 72 °C for 36 s; and a single cycle of denaturation at 95 °C for 15 s, annealing at 60 °C for 1 min, denaturation at 95 °C for 15 s and annealing at 60 °C for 15 s. Data were acquired and analyzed with the 7500 Fast System software version 1.5.1. (Applied Biosystems). Genome copies of each viral segment were finally calculated by intrapolation of the respective standard curve prepared with tenfold serial dilutions of the viral segment cloned in pUC57 plasmids starting at 0.1 ng/μL.

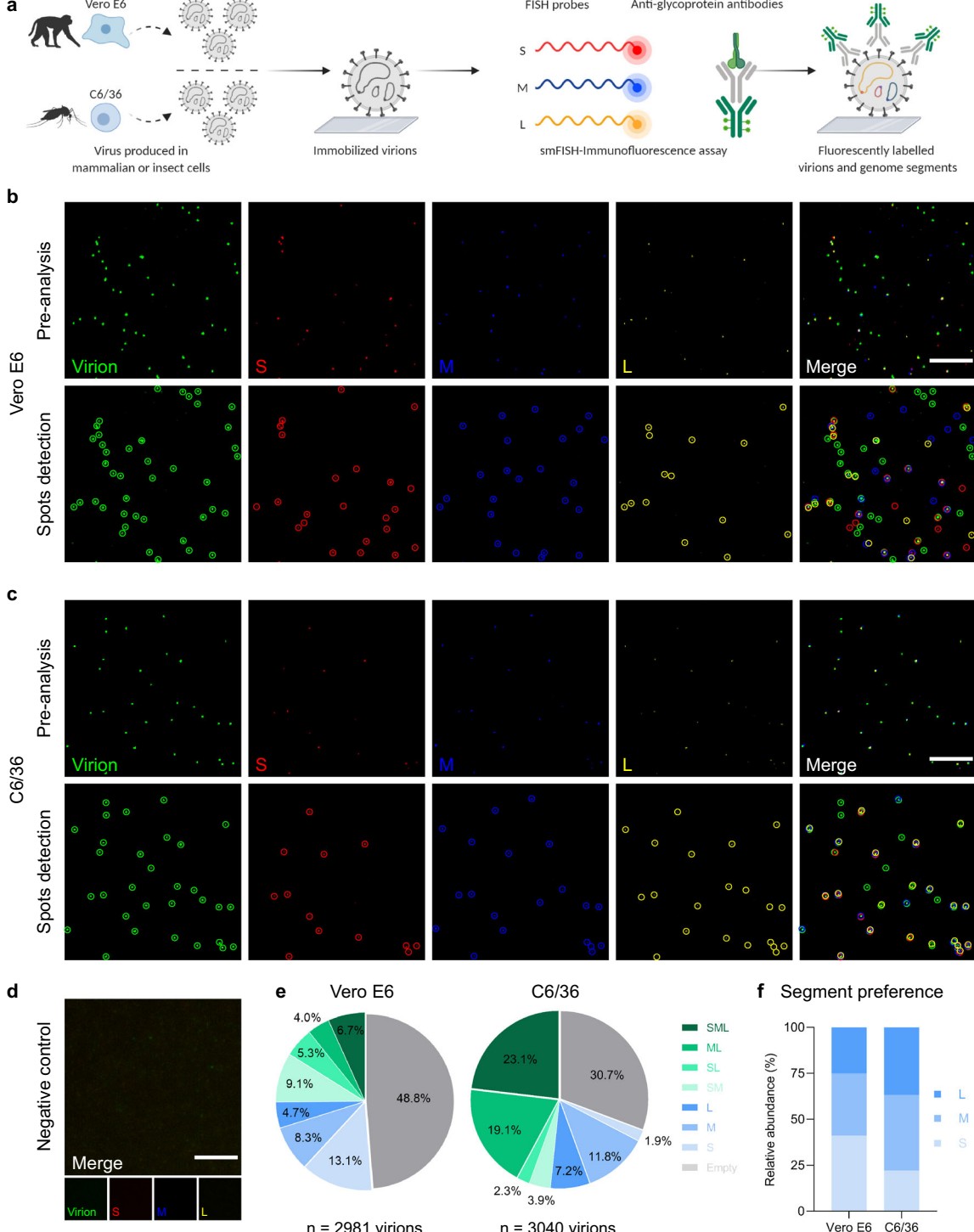

**Fig. 5 Genome segment composition of immobilized RVFV virions produced in mammalian and insect cells. a** Schematic representation of the experimental setup. RVFV virions produced in Vero E6 cells or C6/36 cells were immobilized on coverglass by incubation for 5 h at 28 °C. The S segment (N gene; red), M segment (NSmGn and Gc genes separately; blue) and L segment (RdRp gene; yellow) were hybridized using probe sets labeled with CAL Fluor Red 610, Quasar 670 and Quasar 570, respectively. Progeny RVFV particles (green) were detected with antibody 4-D4[48] targeting the Gn glycoprotein in combination with Alexa Fluor 488-conjugated secondary antibodies. Individual spots, each representing either a single vRNP or a virus particle were detected, counted and assessed for co-localization in ImageJ with the plugin ComDet. Visualization of RVFV virions produced in Vero E6 cells (**b**) or C6/36 cells (**c**) (top rows). Merge images show the overlay of the four individual channels. Colored circles (bottom rows) display the spots detected on each channel and their co-localization in the merge image. Due to a higher fluorescence intensity of the green channel compared to the other channels, spots co-localizing with the glycoprotein may sometimes appear masked and not entirely evident in merged images. **d** Negative control sample using cell culture media instead of a virus stock preparation. Scale bars, 5 μm. **e** Relative abundance of the eight possible genome compositions of the virions produced in Vero E6 cells (left) or C6/36 cells (right). **f** Abundance of each genome segment incorporated into a virion relative to the total genome segment packaging events.

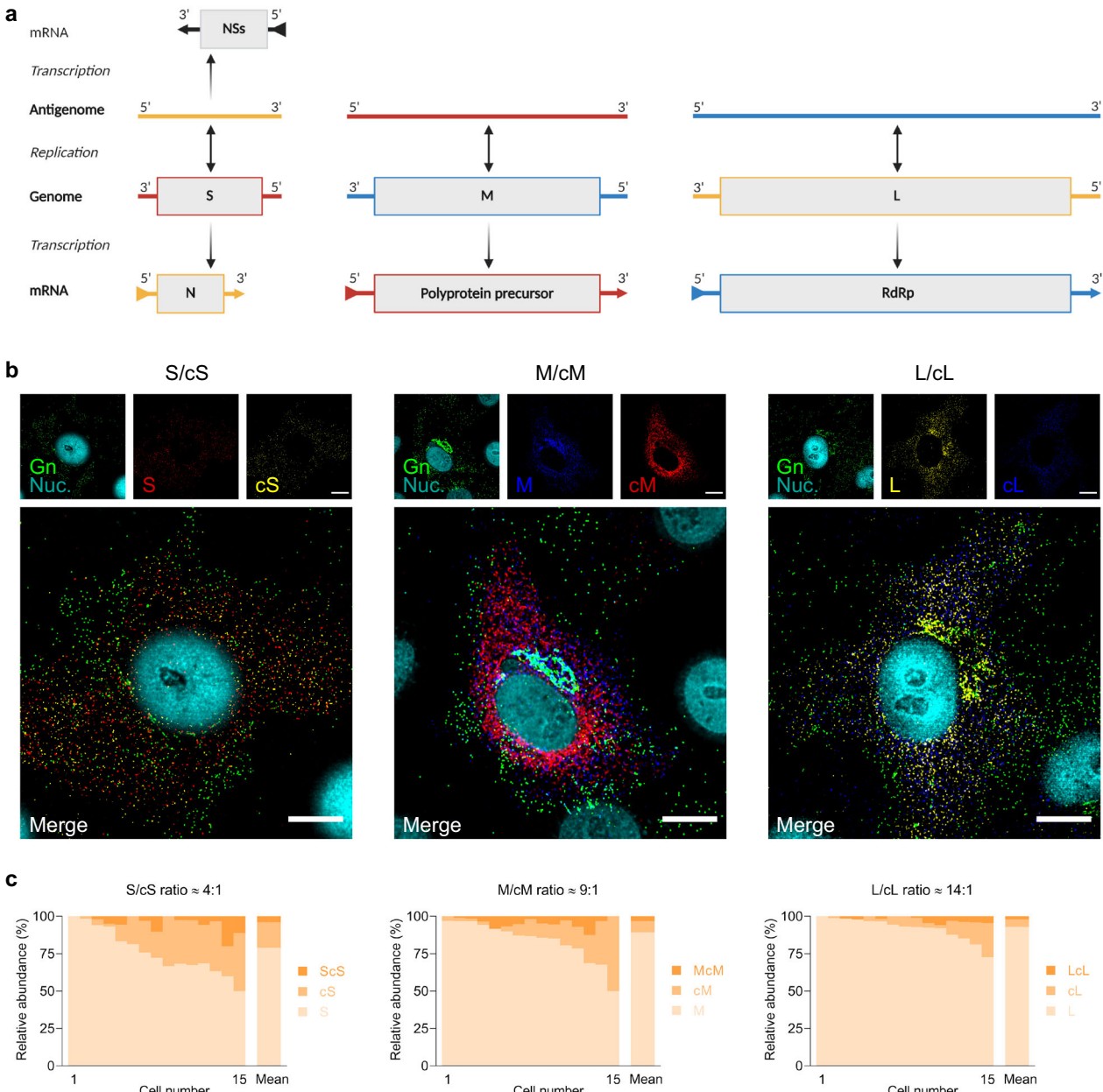

**Fig. 6 Single-molecule RNA FISH-immunofluorescence on vRNA–cRNA pairs of RVFV-infected mammalian cells. a** Schematic representation of the replication and transcription of RVFV genome segments. Here, we refer to viral genome replication intermediates (antigenomes) and mRNA transcripts as cRNAs. RVFV S segment uses an ambisense coding strategy to generate mRNAs from both the genomic-sense and antigenomic-sense RNAs. **b** Visualization of vRNPs, cRNAs and progeny virions of RVFV-infected cells. Vero E6 cells were infected with RVFV (MOI 0.75-1.00) and cells were fixed at 8–10 h post infection. Samples were hybridized against paired targets (i.e., S-cS segments, M-cM segments and L-cL segments). The S segment (N gene; red), M segment (polyprotein gene; blue) and L segment (RdRp gene; yellow) were hybridized using probe sets labeled with CAL Fluor Red 610, Quasar 670 and Quasar 570, respectively. The cS segment (N gene; yellow), cM segment (polyprotein gene; red) and cL segment (RdRp gene; blue) were hybridized using probe sets labeled with Quasar 570, CAL Fluor Red 610 and Quasar 670, respectively. Progeny RVFV particles (green) were detected with antibody 4-D4[48] targeting the Gn glycoprotein in combination with Alexa Fluor 488-conjugated secondary antibodies. Cell nuclei (cyan) were visualized with DAPI. Individual spots, each representing either a vRNP, a cRNA or a virus particle were detected, counted and assessed for co-localization in ImageJ with the plugin ComDet. Main images are merged maximum intensity projections of four channels (individual channels shown on top). Due to a higher fluorescence intensity of the green channel compared to the other channels, spots co-localizing with the glycoprotein may sometimes appear masked and not entirely evident in merged images. Scale bars, 10 μm. **c** Quantification of the S, M, L vRNPs and their corresponding cRNAs in RVFV progeny virions. Genome compositions of the virions are expressed as their abundance relative to the amount of virions in which at least one vRNP or cRNA was detected. Graphs show the composition results of virions released by single cells (*n* = 15 cells per combination; more than 3900 virions per combination) and means. vRNA/cRNA ratios are indicated. cS complementary S segment, cM complementary M segment, cL complementary L segment.

**Virus titration**. Infectious virus titers of samples from the in vitro replication experiments were determined with an immunoperoxidase monolayer assay. Vero E6 cells ($2 \times 10^4$ cells/well) were incubated with tenfold serial dilutions (starting at 1:10) of cell culture supernatants for 72 h at 37 °C and 5% $CO_2$. After incubation, cells were fixed with 4% paraformaldehyde for 15 min, washed with PBS supplemented with 0.5% Tween 80 (PBST), and permeabilized with 1% Triton X-100 in PBS for 5 min. Next, samples were blocked with 100 μL/well of 5% horse serum in PBS and subsequently incubated in sequential steps with 100 μL/well of primary and secondary antibodies. Hybridoma 4-D4[48] supernatant (1:40 dilution) and serum from an experimentally infected sheep (1:1000 dilution), were used as primary antibodies against RVFV and SBV, respectively. As secondary antibodies, HRP-conjugated rabbit polyclonal anti-mouse immunoglobulins (1:500 dilution, Dako) and HRP-conjugated rabbit polyclonal anti-sheep IgG (1:500 dilution, ab6747 Abcam) were used. Incubations with the blocking solution, primary and secondary antibodies were each for 1 h at 37 °C. Plates were washed with PBST between the addition of primary and secondary antibodies. For staining, 100 μL/well of a 0.2 mg/mL amino ethyl carbazole solution in 500 mM acetate buffer pH 5.0, 88 mM $H_2O_2$ was added as substrate. Samples were analyzed in triplicate and the titer calculated as the median tissue culture infectious dose ($TCID_{50}$/mL) using the Spearman–Kärber method. Virus titers of plasma samples were determined with a virus isolation assay as reported[28].

**Single-molecule RNA FISH-immunofluorescence**. Experiments were performed with slight modifications to the Stellaris protocol for simultaneous FISH-immunofluorescence in adherent cells (Biosearch Technologies)[49–51]. Vero E6 cells ($1.5 \times 10^4$ cells/well) or C6/36 cells ($4.5 \times 10^4$ cells/well) were seeded on Culture-Well 16 removable chambered coverglass (Grace Bio-Labs). Following overnight incubation at 37 °C and 5% $CO_2$ (Vero E6) or 28 °C (C6/36), cells were infected with RVFV or SBV at MOIs of 0.33–1.00. One hour post infection, the medium was refreshed. At defined time points (varied per experiment), cells were fixed and permeabilized with a 3:1 mixture of methanol (Merck)—glacial acetic acid (Merck) for 10 min. Cells were subsequently washed twice with PBS and once with pre-hybridization buffer (10% deionized formamide [Millipore] in 2× concentrated SSC [Gibco]) for 5 min. Cells were then incubated for 12–16 h at 37 °C with 100 μL/well of virus-specific FISH probe sets (Supplementary Data 1) and primary antibodies in hybridization buffer (10% deionized formamide, 10% dextran sulfate [Sigma-Aldrich], 2 mM vanadyl ribonucleoside complexes [VRC, Sigma-Aldrich] in 2× SSC). Custom probe sets were designed using the RNA FISH Probe Designer tool (https://www.biosearchtech.com/support/tools/design-software/stellaris-probe-designer) and purchased from Biosearch Technologies (Petaluma, California and Risskov, Denmark). FISH probes were added at a final concentration of 250 nM for RVFV and 125 nM for SBV. Hybridoma 4-D4[48] supernatant (1:160 dilution) and serum from an immunized rabbit[52] (1:4000 dilution), were used as primary antibodies against RVFV and SBV, respectively. Following hybridization and incubation with primary antibodies, cells were extensively washed at 37 °C (twice with pre-hybridization buffer for 30 min and twice with 2× SSC for 15 min). Subsequently, cells were incubated with 100 μL/well of secondary antibodies for 1 h at 37 °C. A goat polyclonal anti-mouse IgG labeled with Alexa Fluor 488 (1:1000 dilution, A-11001 Invitrogen) or a goat polyclonal anti-rabbit IgG labeled with FITC (1:400 dilution, sc-2012 Santa Cruz Biotechnology) were used as secondary antibodies. Next, cells were washed twice with 2× SSC, and nuclei were stained by incubation with 100 μL/well of 1 μg/mL DAPI in 2× SSC for 5 min. Finally, cells were washed with 2× SSC and submerged in VectaShield antifade mounting medium H-1000 (Vector Laboratories). For analysis of virus stocks, 100 μL/well of virus stocks diluted 1:3 were added on CultureWell 16 removable chambered coverglass and virions were allowed to attach to the surface for 5 h at 28 °C. From the fixation step onwards, the same procedure as described for adherent cells was followed. The specificity of the FISH probes and antibodies was confirmed with single-color controls (Supplementary Figs. 1, 2, 6). Mock-infected samples and samples without primary antibodies were used as negative controls.

**Image acquisition and analysis**. Z-stacked images of infected cells and immobilized virions, with a fixed interval of 0.28–0.31 μm between slices, were acquired with an inverted widefield fluorescence microscope Axio Observer 7 (ZEISS, Germany) using appropriate filters and a 1.3 NA 100x EC Plan-NEOFLUAR oil objective in combination with an AxioCam MRm CCD camera. Exposure times were defined empirically and differed depending on the cell line, probe sets and fluorescent dyes. Raw images were deconvolved in standard mode using Huygens Professional version 19.10 (Scientific Volume Imaging B.V., The Netherlands). If required, raw images were Z-aligned in ZEN 2.6 Pro (ZEISS, Germany) before deconvolution. For analysis, 3D data were converted to maximum intensity projections using Z-project within ImageJ[53]. Detection, quantification and co-localization analyses of individual spots, each representing a single virion, vRNP or cRNA, were performed in ImageJ in combination with the plugin ComDet version 0.5.0 (https://github.com/ekatrukha/ComDet). Spot detection thresholds for each channel were set empirically by individual examination of images. The threshold to define co-localized spots was set to a maximum distance of 3–4 pixels between the centers of the spots. Intracellular genome composition analysis considered a region of the cytoplasm representative of the overall composition, not including the Golgi apparatus where signal is generally overcrowded due to vRNP accumulation (Fig. 3d and Supplementary Fig. 3). Genome composition analysis of progeny

virions only considered virus particles in selected regions of interest located distant from the nucleus of the infected cell (Fig. 3e and Supplementary Fig. 3). For visualization purposes, image brightness and contrast were manually adjusted in ImageJ. Finally, Imaris 9.5 software (Bitplane, Switzerland) was utilized to create optimal 3D representations of the data using the Surface and Spots modes.

**Statistics and reproducibility**. Prism 8 (GraphPad Software) was used to generate graphs and perform statistical analysis. Sample size varied per experiment and is indicated in each figure legend. Mean vRNA:infectivity ratios were compared using an unpaired two-tailed Student's $t$ test with Welch's correction (not assuming equal variances). The correlation between the intracellular vRNP relative frequency and packaged vRNP relative frequency was calculated with the Pearson's correlation coefficient (r). $p$ values ≥ 0.05 were considered not significant.

**Ethics statement**. The animal experiment within the scope of another study[28] from which plasma samples were obtained for analysis was conducted in accordance with European regulations (EU directive 2010/63/EU) and the Dutch Law on Animal Experiments (Wod, ID number BWBR0003081). Permissions were granted by the Dutch Central Authority for Scientific Procedures on Animals (Permit Number: AVD4010020185564). Specific procedures were approved by the Animal Ethics Committees of Wageningen Research.

**Reporting summary**. Further information on research design is available in the Nature Research Reporting Summary linked to this article.

## Data availability
The authors declare that the data supporting the findings of this study are available within the paper and its supplementary information files. Source data underlying Figs. 1b–i, 1k–m, 2f, g, 4a–e, 5e, f and 6c are provided in the Supplementary Data 2 file. Any remaining data are available from the corresponding author upon reasonable request.

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

## Acknowledgements

E.B.M. is a grateful recipient of scholarships from the Graduate School for Production Ecology & Resource Conservation (PE&RC) and Universidad de Costa Rica (OAICE-031-2019). We thank Corinne Geertsema (Laboratory of Virology, Wageningen University) for kindly providing the KC cells, Michèle Bouloy (Institut Pasteur, France) for kindly providing the RVFV strain Clone 13, Eduardo de Freitas Costa and Jose L. Gonzalez (Department of Epidemiology, Bioinformatics and Animal Studies, Wageningen Bioveterinary Research) for advice on statistics, Hélène Cecilia (UMR 1300 BIOEPAR, INRAE/Oniris) and Quirine ten Bosch (Quantitative Veterinary Epidemiology, Wageningen University) for stimulating discussions, and Richard Kormelink (Laboratory of Virology, Wageningen University) for critically reviewing the manuscript and providing feedback. Illustrations in Figs. 1a, j, 2a, 3a, f, 4f, 5a and 6a were created with BioRender.com.

## Author contributions

P.J.W.S. and J.K. conceived the project. C.M.S. contributed developing the 5-channel FISH-immunofluorescence method. E.B.M. and P.J.W.S. designed the experiments. E.B.M. performed the RT-qPCR, virus titration and FISH-immunofluorescence experiments. E.A.K. developed and customized the plugin ComDet for image analysis. E.B.M. and P.J.W.S. analyzed and interpreted the data with the contributions of J.K. and E.A.K. P.J.W.S. and J.K. supervised the project. E.B.M and P.J.W.S. wrote the manuscript, prepared the figures and movies with the contributions of J.K.

## Competing interests

The authors declare no competing interests.
