## [Peer Review File · Communications Biology]

Referee expertise:

Referee #1: viral genome segment fluorescence microscopy

Referee #2: viral genome assembly, FISH

Reviewers' comments:

Reviewer #1 (Remarks to the Author):

The authors present original experimental data suggesting that bunyavirus genome packaging varies in a cell-specific manner. They report more efficient packaging in insect cells compared to mammalian cells. In addition, by using quantitative microscopy, the authors report that RNA genome packaging in bunyaviruses is influenced by the intracellular viral RNA composition (i.e., 'balanced'- close to equimolar ratios for all three RNA segments vs 'imbalanced'). These findings are original and they authors are building up on their previous extensive studies of bunyavirus stochastic genome packaging. Understanding of the spread of RNA viruses with segmented genomes is an important area of research, and currently very little is known about the spread of viruses producing multiple particles with incomplete genomes. Thus, this novel study provides new insights into the distribution of RNA segments in virus-infected cells, and how this distribution may affect genome packaging efficiencies. Unfortunately, it does not provide any mechanistic insights into why and how the observed differences in the intracellular distribution of RNAs could alter genome packaging efficiencies. The authors mention an interesting observation that the number of RVFV genomic copies is more than 10 times higher in Vero cells than in C6/36 cells – could the growth temperature for different cells be an important contributing factor to higher initiation/elongation rates for the viral polymerase, resulting in the higher copy numbers in mammalian cells, despite similar resulting infectious titers? Could the authors alter the RNA packaging balance by over-expressing one of the segments in virus-infected cells – would this affect packaging efficiencies? Overall, this is an interesting study, I have several suggestions and queries with respect to their results and data interpretation.

Specific comments:

1. Lines 110-116 – I am not sure how relevant the mentioned study is in this context. The authors quantified RVFV genome copies in plasma samples, implying that genome packaging efficiencies in lambs vary over time. They only report measuring the infectious virus and RNA copies by RT-PCR – I think from these data one can only judge the presence of circulating RNAs, but not directly assess packaging efficiencies. I felt this study was rather out of place (or perhaps it wasn't fully explained).
2. Line 130-132 – the observed heterogeneity between cells – I think the correct interpretation would be not just intrinsic cell-to-cell variability within the population, but rather different number of particles entering cells, some of which would be fully infectious, and some missing some genomic segments, as the study suggest. I would thus rectify this claim.
3. Line 195 – This claim should be also clarified, as the data does not rule out the possibility that the imbalanced intracellular composition is not the corollary of the initially imbalanced infection (more particles with missing segments infecting the same cell). To rule this out, one would need to do a single virus infection experiment, or infect the cells with a purified, highly infectious virus stock with most particles containing a complete set of segments. To this end, can the authors comment on the proportion of particles containing all segments in their inoculum used for infecting cells?
4. Figs. 2& 3 – I don't think it is posible to detect a single virion particle using conventional

immunofluorescence with FITC-conjugated antibodies for SBV –(or Alexa Fluor488 for RVFV that matter). While the authors demonstrate that smFISH does detect single viral RNAs, the IF data showing single virions is not convincing. Using this detection approach and conventional diffraction limited microscopy one cannot resolve single virus particles and distinguish them from the viral proteins that accumulate in infected cells.

5. Fig. 4 - for virus particle immobilization – how were these particles purified? Are these density gradient purified viruses, i.e., no viral extracellular RNAs that would be obvious on the glass surface. There are RNA-only spots shown in the combined panel (merge), not colocalising with the green channel), suggesting that the FISH probes non-specifically interacted with something immobilized on the cover glass. More details would be helpful to understand how this experiment was performed.

Reviewer #2 (Remarks to the Author):

The manuscript by Bermúdez-Méndez et al. describes packaging of viral ribonucleocomplexes (vRNPs) by bunyaviruses using combined fluorescence in situ hybridization (FISH) and immunofluorescence. The study compares vRNP packaging efficiencies of two bunyaviruses, Rift Valley fever virus (RVFV) and Schmallenberg virus (SBV), in mammalian and mosquito cells. This study demonstrates that the ratio of vRNA to infectious virus increases over time in mammalian but not mosquito cells. The authors go on to show that the fraction of bunyavirus virions, defined as Gc- or Gn-positive foci, that package vRNPs is only 35-55% in mammalian cells, with as few as 10% of virions packaging all three (S, M and L) vRNP segments. The authors go on to show that vRNP packaging efficiency in single cells is heterogeneous, with cells expressing equal ratios of each vRNP segment being more likely to package all three vRNP segments. Interestingly, vRNP packaging efficiency was higher in insect cells, which correlates with their initial findings that the ratio of vRNA to infectious virus particles is lower in insect cells than in mammalian cells. Lastly, the authors show that packaging of complementary RNA (cRNA) occurs at a low frequency, suggesting that vRNP packaging is less selective in bunyaviruses. Overall, the study is well-done and contributes several interesting new findings to the bunyavirus field. Addressing the following points would enhance the manuscript:

1. The authors define bunyavirus virions in this study as any Gc- or Gn-positive foci. This is problematic, as their analysis could capture both mature virions and immature Gc or Gn protein that has yet to undergo budding. This distinction could explain why as many as half of the Gc-/Gn-containing foci do not colocalize with the S, L or M segments (Fig. 4a). This possibility should be discussed in the text.
2. The Methods should clearly state the distance threshold for defining colocalization of individual spots.
3. The fraction of Gc/Gn-containing foci that packages any vRNP segment in mammalian cells is only 35-55% at 8 hpi. However, the vRNA to infectious virus ratio in RVFV infection goes up both in mammalian cells in vitro and in the RVFV-infected lambs in vivo (Fig. 1h & 1m). Do the authors have any data on vRNP packaging from other time points to determine whether vRNP packaging efficiency changes over time?
4. The authors describe the packaging efficiency of RVFV and SBV in mammalian cells as roughly equivalent (lines 158-161), yet the packaging efficiency of vRNPs is generally higher following SBV infection (Fig. 4a). This should be clarified in the text.
5. The finding that vRNP packaging is more efficient in insect cells is intriguing. The authors demonstrate that the proportion of 'empty' virions decreases in insect cells as compared to mammalian cells (Fig. 5d). This decrease is associated with a concomitant increase in SML and ML virions in insect cells as well as a decrease in SL and SM virions (Fig. 5d), suggesting that packaging of the M and L segments together is more efficient than packaging of the S segment with either of the other two. This also seems to be supported by the fact that the relative abundance of the S segment is

much lower in insect cells than in mammalian cells (Fig. 5e). How does this compare to colocalization of each vRNP segment in the cytoplasm of RVFV-infected mammalian cells, where the S segment is more abundant than the M segment? Or in SBV-infected cells, where the ratio of the S, L and M segments is nearly 1:1:1?

Houtribweg 39, 8221 RA Lelystad, The Netherlands

02 November 2020

Response to reviewers letter (COMMSBIO-20-2323-T)

Dear Reviewers,

We would like to thank you for your evaluations, constructive comments and suggestions to improve our manuscript entitled “*Visualizing the RNP content of single bunyavirus virions reveals more efficient genome packaging in the arthropod host*”. In this letter, we provide a point-by-point response to all the comments you raised. In the revised manuscript, changes to the initial submitted version appear highlighted in red.

We hope to have answered your questions and addressed your comments, ultimately refining our manuscript to become acceptable for publication in *Communications Biology*.

Sincerely yours,

Dr. Paul J. Wichgers Schreur

Senior Scientist | Department of Virology, Wageningen Bioveterinary Research

Email: paul.wichgersschreur@wur.nl

Reviewers' comments:

Reviewer #1 (Remarks to the Author):

The authors present original experimental data suggesting that bunyavirus genome packaging varies in a cell-specific manner. They report more efficient packaging in insect cells compared to mammalian cells. In addition, by using quantitative microscopy, the authors report that RNA genome packaging in bunyaviruses is influenced by the intracellular viral RNA composition (i.e., 'balanced'- close to equimolar ratios for all three RNA segments vs 'imbalanced'). These findings are original and they authors are building up on their previous extensive studies of bunyavirus stochastic genome packaging. Understanding of the spread of RNA viruses with segmented genomes is an important area of research, and currently very little is known about the spread of viruses producing multiple particles with incomplete genomes. Thus, this novel study provides new insights into the distribution of RNA segments in virus-infected cells, and how this distribution may affect genome packaging efficiencies. Unfortunately, it does not provide any mechanistic insights into why and how the observed differences in the intracellular distribution of RNAs could alter genome packaging efficiencies. The authors mention an interesting observation that the number of RVFV genomic copies is more than 10 times higher in Vero cells than in C6/36 cells – could the growth temperature for different cells be an important contributing factor to higher initiation/elongation rates for the viral polymerase, resulting in the higher copy numbers in mammalian cells, despite similar resulting infectious titers? Could the authors alter the RNA packaging balance by over-expressing one of the segments in virus-infected cells – would this affect packaging efficiencies? Overall, this is an interesting study, I have several suggestions and queries with respect to their results and data interpretation.

Response: We thank the reviewer for his/her thorough analysis and assessment of the paper and take it as a compliment that the results were found interesting. We agree that the detailed mechanisms explaining the differences in genome packaging efficiencies between hosts are not yet unravelled. Nevertheless, we believe this study provides an important starting point for follow up studies focusing on RNP transport and virion assembly. As pointed out by the reviewer, temperature could be a factor influencing bunyavirus genome packaging by for instance affecting polymerase activity, thereby explaining the higher RNA copy numbers in mammalian cells. However, increased polymerase activity is likely to indirectly result in increased expression of viral proteins and subsequent assembly of virions, potentially incorporating the additional RNPs synthesized. Furthermore, temperature may also affect other packaging-related processes, such as RNP transport to the site of virion assembly. Hence, a fair evaluation of the role of polymerase activity is complex.

Overproducing one of the segments could indeed be an interesting approach. However, by overproducing a specific genome segment, besides skewing the intracellular vRNP content towards that specific segment, we would also change the overall viral protein balance and consequently would interfere with other factors likely involved in the genome packaging process. In spite of that, we want to thank the reviewer for these interesting thoughts and suggestions.

Specific comments:

1. *Lines 110-116 – I am not sure how relevant the mentioned study is in this context. The authors quantified RVFV genome copies in plasma samples, implying that genome packaging efficiencies in lambs vary over time. They only report measuring the infectious virus and RNA copies by RT-PCR – I think from these data one can only judge the presence of circulating RNAs, but not directly assess packaging efficiencies. I felt this study was rather out of place (or perhaps it wasn't fully explained).*

Response: We acknowledge the fact that the data from plasma samples were not clearly introduced within the text and that these data only provide indirect evidence regarding packaging efficiencies. We have tried to assess the RNP content in virions present in plasma, however immobilizing virions from the plasma matrix appeared very challenging and did so far not result in quantifiable data. Nevertheless, we consider that the increase of vRNA:infectivity ratio over time in an *in vivo* setting provides new valuable information that points toward heterogeneity in packaging efficiency in time, in addition to heterogeneity in packaging efficiencies between hosts. In order to improve the clarity of these results, we have introduced in the Results section (lines 113-114) an explanatory phrase to contextualize the aim of analyzing the plasma samples.

2. *Line 130-132 – the observed heterogeneity between cells – I think the correct interpretation would be not just intrinsic cell-to-cell variability within the population, but rather different number of particles entering cells, some of which would be fully infectious, and some missing some genomic segments, as the study suggest. I would thus rectify this claim.*

In this specific passage of the text, we are exclusively referring to the observed glycoprotein staining for detecting virions (Fig. 2). Synthesis of the glycoproteins encoded by the M segment can only occur in cells infected with a complete set of genome segments due to the requirement of the N protein and the viral polymerase for viral genome replication and transcription. Thus, we sustain that the observed variability in the kinetics of glycoprotein synthesis and virion assembly is likely a consequence of intrinsic cell-to-cell variability regarding virus entry and viral gene expression. To clarify that we specifically refer to our observation on glycoprotein levels and number of progeny virions, we have introduced an additional phrase in that passage (lines 132-134).

3. *Line 195 – This claim should be also clarified, as the data does not rule out the possibility that the imbalanced intracellular composition is not the corollary of the initially imbalanced infection (more particles with missing segments infecting the same cell). To rule this out, one would need to do a single virus infection experiment, or infect the cells with a purified, highly infectious virus stock with most particles containing a complete set of segments. To this end, can the authors comment on the proportion of particles containing all segments in their inoculum used for infecting cells?*

Response: We acknowledge the possibility that an imbalanced intracellular composition could, at least partly, be the consequence of infection by a combination of complete and incomplete particles. We also agree with the reviewer that a single virion infection experiment or an infection with a virus stock preparation with particles containing almost exclusively a complete set of genome segments is the best way to ensure that a single cell has only been infected with one infectious virus particle, as the initial vRNP content would be balanced. However, such an experimental setup is technically very complex due to the challenge of sorting individual virus particles and subsequent selection and purification of particles with a defined genome content. Furthermore, we want to note that we reduced the possibility of co-infection by multiple virions by infecting cells at a multiplicity of infection (MOI) ranging from 0.33-1.00. The inoculum used was produced in Vero E6 cells and corresponds to the virus stock preparation analyzed in Fig. 5e (approximately 7% of the particles contained the three genome segments). To point out that indeed, an imbalanced intracellular composition could be influenced by multiple factors including co-infection of incomplete particles, we have clarified this in the Discussion section (lines 261-264).

4. *Figs. 2& 3 – I don't think it is possible to detect a single virion particle using conventional immunofluorescence with FITC-conjugated antibodies for SBV –(or Alexa Fluor488 for RVFV that matter). While the authors demonstrate that smFISH does detect single viral RNAs, the IF data showing single virions is not convincing. Using this detection approach and conventional diffraction limited microscopy one cannot resolve single virus particles and distinguish them from the viral proteins that accumulate in infected cells.*

Response: We appreciate the concern raised about the method we used to detect individual virus particles. However, the following reasons support our claim: smFISH detection of individual RNA molecules is based on the collective signal generated by 25-48 fluorescent probes that bind to different regions of a single RNA molecule. In our detection of single virus particles (i) the primary antibodies used recognize the viral glycoproteins, that in the context of an infected cell, start to accumulate in the Golgi apparatus (the site of virion assembly) and are then displayed at the surface of virions. By tracking infected cells over time, we observed glycoprotein staining only after accumulation of the glycoproteins

in a perinuclear region at 5 ± 1 hour post-infection, and accumulation of signal in the perinuclear region plus disperse individual spots at a later stage (7 ± 1 hours post-infection and later). Both the accumulation of signal in a perinuclear region and the later appearance of disperse individual spots are in line with the bunyavirus (three-segmented) replication cycle kinetics described so far, in which newly formed virions bud solely from the trans-Golgi network and can be detected around 6-8 hours post-infection. (ii) Moreover, the signal from each individual spot is generated by numerous antibodies that cover the surface of each mature particle. For each RVFV particle containing 720 Gn molecules, 720 potential Gn antibody binding sites are available (Huiskonen et al., J Virol. 2009 Apr;83(8):3762-9) and probably similarly numerous Gc binding sites are available on a SBV particle. (iii) Single-particle detection is characterized by an unimodal distribution of fluorescence intensity along a line crossing a spot, as shown in Fig. 2f. The area of the detected spots also depicted an unimodal distribution, denoting reproducible measurements of single spots within and between images (Fig. 2g). (iv) With the aim of exclusively analyzing mature individual virus particles and to discriminate them from viral glycoproteins accumulating in the cell, we only took into account in our analysis individual spots located relatively far away from the nucleus, most likely from the extracellular space. Example regions of interest analyzed are depicted in Fig. 3e and Supplementary Fig. 3. With this, we hope we have convinced the reviewer that our methodology enables the visualization of individual particles.

5. Fig. 4 - for virus particle immobilization – how were these particles purified? Are these density gradient purified viruses, i.e., no viral extracellular RNAs that would be obvious on the glass surface. There are RNA-only spots shown in the combined panel (merge), not colocalising with the green channel), suggesting that the FISH probes non-specifically interacted with something immobilized on the cover glass. More details would be helpful to understand how this experiment was performed.

Response: We are assuming the reviewer refers to Fig. 5 instead of Fig. 4. For this experiment, virions were analyzed directly from virus stock preparations in which the culture supernatant of infected cells was only clarified by centrifugation to remove cell debris. No additional purification procedures were employed to prevent any bias in the particle population. For this reason, it makes sense that we also detected some RNA-only spots. To rule out that the FISH probes did not interact non-specifically with the cover glass we always took along negative controls. In the revised manuscript we have now modified Fig. 5 to include images of the negative control of this experiment (Fig. 5d). Details were also included in the figure legend (line 703).

Reviewer #2 (Remarks to the Author):

The manuscript by Bermúdez-Méndez et al. describes packaging of viral ribonucleocomplexes (vRNPs) by bunyaviruses using combined fluorescence in situ hybridization (FISH) and immunofluorescence. The study compares vRNP packaging efficiencies of two bunyaviruses, Rift Valley fever virus (RVFV) and Schmallenberg virus (SBV), in mammalian and mosquito cells. This study demonstrates that the ratio of vRNA to infectious virus increases over time in mammalian but not mosquito cells. The authors go on to show that the fraction of bunyavirus virions, defined as Gc- or Gn-positive foci, that package vRNPs is only 35-55% in mammalian cells, with as few as 10% of virions packaging all three (S, M and L) vRNP segments. The authors go on to show that vRNP packaging efficiency in single cells is heterogeneous, with cells expressing equal ratios of each vRNP segment being more likely to package all three vRNP segments. Interestingly, vRNP packaging efficiency was higher in insect cells, which correlates with their initial findings that the ratio of vRNA to infectious virus particles is lower in insect cells than in mammalian cells. Lastly, the authors show that packaging of complementary RNA (cRNA) occurs at a low frequency, suggesting that vRNP packaging is less selective in bunyaviruses. Overall, the study is well-done and contributes several interesting new findings to the bunyavirus field. Addressing the following points would enhance the manuscript:

1. The authors define bunyavirus virions in this study as any Gc- or Gn-positive foci. This is problematic, as their analysis could capture both mature virions and immature Gc or Gn protein that has yet to undergo budding. This distinction could explain why as many as half of the Gc-/Gn-containing foci do not colocalize with the S, L or M segments (Fig. 4a). This possibility should be discussed in the text.

Response: Staining with a primary antibody recognizing Gn (RVFV) or Gc (SBV) indeed detects both mature virions as well as immature glycoprotein not yet displayed on the virion surface. However, by following the kinetics of infection, we show that at early stages of infection, glycoprotein staining is only detected as an accumulated signal in a perinuclear region (immature glycoprotein in the Golgi apparatus, the site of virion assembly), whereas at later stages of infection, besides the signal accumulation in the perinuclear region, we observed disperse individual spots, highly likely to represent mature virions (Fig. 2b-e). To avoid incorrectly detecting immature glycoprotein as mature virions, in our analysis we exclusively selected individual Gn or Gc spots located relatively far away from the nucleus, most likely from the extracellular space. Example regions of interest analyzed are depicted in Fig. 3e and Supplementary Fig. 3. Finally, it is worth mentioning that the fraction of RVFV empty particles from both infected cells and the virus stock preparation is very similar (close to 50%). We would expect that if immature particles were mistakenly taken into account in the analysis of infected cells, the fraction of empty particles from these samples would be considerably larger. We hope that with the above explanation and supporting images referred in the figures we have convinced the

reviewer that our analysis of vRNP content in virions is based on data from mature virus particles and not immature glycoprotein.

2. The Methods should clearly state the distance threshold for defining colocalization of individual spots.

Response: We agree with the reviewer and have added in the Methods section (lines 443-444) the distance threshold for defining colocalization of single spots.

3. The fraction of Gc/Gn-containing foci that packages any vRNP segment in mammalian cells is only 35-55% at 8 hpi. However, the vRNA to infectious virus ratio in RVFV infection goes up both in mammalian cells in vitro and in the RVFV-infected lambs in vivo (Fig. 1h & 1m). Do the authors have any data on vRNP packaging from other time points to determine whether vRNP packaging efficiency changes over time?

Response: We agree with the reviewer that having additional data on vRNP packaging efficiencies at later time points would be very informative and could change over time. However, carrying out the analysis as presented in Figs. 3 and 4 at later time points is, at least with the current microscope setup and resolution, not possible due to very high densities of intracellular vRNPs, resulting in accumulated signal throughout the whole cytoplasm and loss of single-molecule resolution. Moreover, as explained in our response to specific comment #1 of reviewer #1, we did not succeed in assessing RNP content of virions present in plasma. We now refer to this matter in the Discussion section (lines 300-306).

4. The authors describe the packaging efficiency of RVFV and SBV in mammalian cells as roughly equivalent (lines 158-161), yet the packaging efficiency of vRNPs is generally higher following SBV infection (Fig. 4a). This should be clarified in the text.

Response: The observation of the reviewer is fully accurate. We have now clarified the differences between packaging efficiency of RVFV and SBV in the Results section (lines 167-169).

5. The finding that vRNP packaging is more efficient in insect cells is intriguing. The authors demonstrate that the proportion of 'empty' virions decreases in insect cells as compared to mammalian cells (Fig. 5d). This decrease is associated with a concomitant increase in SML and ML virions in insect cells as well as a decrease in SL and SM virions (Fig. 5d), suggesting that packaging of the M and L segments together is more efficient than packaging of the S segment with either of the other two. This also seems to be supported by the fact that the relative abundance of the S segment is much lower in insect cells than in mammalian cells (Fig. 5e). How does this compare to colocalization of each vRNP segment in the cytoplasm of RVFV-infected mammalian cells, where the S segment is more abundant

than the M segment? Or in SBV-infected cells, where the ratio of the S, L and M segments is nearly 1:1:1?

Response: In line with the correlation observed between the relative frequencies of intracellular vRNPs and the frequency of packaged vRNPs (Fig. 4d), we postulate that the higher abundance of the S segment compared to the M segment in RVFV-infected mammalian cells leads to the S segment being incorporated into particles more often than the M segment (Fig. 4b, c). In the case of SBV-infected mammalian cells, where there is on average a nearly 1:1:1 intracellular content of vRNPs, as noted in our proposed model on the efficiency of genome packaging (Fig. 4f), it was more common to observe cells with a relative efficient packaging compared to RVFV-infected cells (Fig. 4c). However, as can be drawn from the data, a balanced intracellular vRNP content does not always ensure an efficient packaging, which may explain the slightly different frequencies at which the S, M and L segments were incorporated into mature virions.

REVIEWERS' COMMENTS:

Reviewer #2 (Remarks to the Author):

The authors have appropriately addressed all of the original comments.